# Caregivers’ Experience of Supporting Deaf Adults with Mental Health Disorders in Ghana

**DOI:** 10.3390/ijerph22020144

**Published:** 2025-01-22

**Authors:** Wisdom Kwadwo Mprah, Maxwell Peprah Opoku, Ebenezer Mensah Gyimah, Shakila Nur, Juventus Duorinaah, Lilian Frimpomaa, Maria Efstratopoulou

**Affiliations:** 1Centre for Disability and Rehabilitation Studies, Department of Health Promotion and Disability Studies, Kwame Nkrumah University of Science and Technology, PMB, KNUST, Kumasi AK-448-4944, Ghana; wmprah.chs@knust.edu.gh (W.K.M.); gyimahebenezermensah@gmail.com (E.M.G.); 2Special Education Department, College of Education, United Arab Emirates University, Al-Ain P.O. Box 15551, United Arab Emirates; maria.efstratopoulou@uaeu.ac.ae; 3Department of English and Modern Languages, North South University, Dhaka 1229, Bangladesh; shakila.nur@northsouth.edu; 4Ghana National Association of the Deaf, P.O. Box AN-7908, Accra GA-44-35-222, Ghana; juventusdnaah@yahoo.com; 5Department of Sociology and Social Work, Kwame Nkrumah University of Science and Technology, PMB, KNUST, Kumasi AK-448-4944, Ghana; lillianfrimpomaa400@gmail.com

**Keywords:** mental health, disability, deaf, caregivers, Ghana

## Abstract

Background: Caregivers play a crucial role in the support and management of individuals experiencing mental health disorders; however, there is a paucity of research concerning the experiences of caregivers of deaf persons diagnosed with any form of mental disorder in Ghana. This study aims to explore the lived experiences of these caregivers, with a particular focus on their perceptions of mental health, available support systems, challenges encountered, and the consequent impact on familial relationships while supporting this underserved population. Method: This qualitative study engaged eleven family caregivers of deaf adults diagnosed with a mental health disorders made up of ten females and one male aged 45–68 years, recruited across four of the sixteen administrative regions in Ghana. Utilizing a semi-structured interview guide, face-to-face interviews were conducted to gather in-depth narratives from the participants. Results: Thematic analysis of the data revealed several key themes, including the awareness and understanding of mental health disorders, availability and accessibility of mental health support services and training, dynamics of psychosocial and familial relationships, and the multifaceted challenges faced in caregiving for individuals with mental health disorders. Conclusion: The findings underscored an urgent need for the development of a specialized caregiving manual tailored for caregivers of deaf individuals experiencing mental health disorders. This study advocates for policymakers to prioritize the creation of such resources to enhance care delivery and improve overall mental health outcomes for this vulnerable population.

## 1. Introduction

Globally, mental health disorders have been described as a public health menace due to their effect on society [1,2]. In this study, a mental health disorder was operationalized as a significant disturbance in one’s ability to function cognitively and emotionally [3]. Recent estimates by Effatpanah et al. [4] indicate that 970 million individuals are affected by various mental health disorders, with anxiety, depression, schizophrenia, and bipolar disorder being the most prevalent [5]. The adage “a healthy mind resides in a healthy body” underscores the importance of societal engagement in understanding the prevalence of mental health disorders and in developing frameworks that enable individuals to realize their innate potential. In response, there is a pressing call for governments to formulate comprehensive policies aimed at alleviating the burdens associated with mental health while promoting overall individual well-being [6]. Consequently, governments have been urged to develop robust policies that are aimed towards alleviating mental health burdens while promoting individuals’ well-being [6]. However, in many non-Western contexts, including Ghana, mental health disorders have not been prioritized by policymakers [7,8]. In view of this, persons with mental health disorders are more likely to be undiagnosed or receive inadequate healthcare [7,9,10]. This situation disproportionately affects marginalized populations, who already face substantial barriers to accessing basic health services [11]. In light of these challenges, it is important to explore the experiences of caregivers supporting individuals with mental health disorders in Ghana, especially vulnerable populations such as deaf persons.

Deafness is an audiological condition, a disability, denoting partial or total hearing loss, which interferes with their day-to-day living experiences [11,12]. It draws on the perception that malformation of the auditory nerves contributes to the inability of individuals to receive information and respond accordingly. This interruption in the sensory processing affects the ability of such individuals to engage in daily living activities without appropriate accommodations. One unique characteristics of deaf persons is their inability to communicate orally and thus, their reliance on alternative and augmentative communication such as sign language, visualization, cochlear implants, and hearing aids to interact with the larger society [13,14,15]. Additionally, deafness is defined within the context of cultural diversity and draws parallels between deaf person’s experiences and the experiences of other minorities, such as Blacks and Latinos, share common characteristics which are their unique mode of communication and experiences [16,17]. From this perspective, deaf persons see non-deaf persons as individuals who lack an understanding of their mode of communication and culture [18]. As the larger society is epitomized by oral communication, the Thus, besides their hearing loss, their distinct cultural and linguistic characteristics which include their unique mode of communication differentiates them from other members of society. Unfortunately, in many societies, sign language is only known to members of the deaf community or close family members and sign language interpreters [19,20]. Consequently, deaf persons struggle to participate in societal activities such as healthcare and education.

Deaf persons face unique barriers in their efforts to access healthcare, including inadequate provision of sign language interpretation services and lack of sign language proficiency among healthcare providers [20], which often led to misdiagnoses [21,22]. Additionally, there are limited culturally competent health professionals resulting in outright exclusion from or poor access to health programmes, such as Ghana’s National Health Insurance Scheme, targeting vulnerable groups [23,24]. Moreover, stigmatization against the deaf community, which is rooted in traditional beliefs, including associations with curses, usually discourages care-seeking among deaf people [25,26]. The low health literacy, likewise hinders navigation of the health care system, exacerbating the underutilization of important health care services, especially for rural and uneducated deaf persons [27]. In terms of access to mental health services, there is evidence of limited knowledge among deaf persons when it comes to choosing the appropriate health care services [11,28]. There is also evidence that many deaf persons rely especially on caregivers for information and support services [19,29,30], some of whom do not have adequate knowledge on mental health disorders, support services available and how to care for deaf persons with mental health disorders. Although research on mental health services for deaf persons is still in its embryonic stage [11,30], it is useful to understand the perspectives of their caregivers when it comes to supporting deaf persons with mental health disorders.

Caregivers hold a pivotal position in the lives of individuals experiencing mental health challenges [31,32,33,34]. In the current investigation, caregivers were identified as family members who provide unpaid support to their relatives with mental health disorders. The World Health Organization (WHO) emphasizes the crucial role that caregivers play in facilitating the recovery of those with medical conditions, particularly in the realm of mental health [35,36]. The role of caregivers is synonymous with early recovery, easing the mental health burden and enhancing the recovery of their members who are battling medical conditions [31,32,33,34]. In non-Western contexts, it is reported that approximately 90% of caregivers are family members who engage in support activities such as administering medications, preparing meals, providing supervision, accompanying patients to medical appointments, and offering financial assistance. In Ghana, caregiving is intertwined with the local culture [37]. For instance, individuals have a responsibility to care for one another, and in the face of illness or adversity familial obligations necessitate the support of affected members [37]. Indeed, studies conducted in Ghana and similar contexts indicated that caregivers have a substantial impact on the lives of their family members with mental health disorders [37,38,39]. Notwithstanding this, other studies have also reported significant positive experiences such as resilience [40], an enhanced bonding effect [32], and receiving social support from neighbors [33]. Further, the contextual background of caregivers has been identified as an influence on the caregiving experience, with caregivers of Asian and minority ethnic backgrounds being more likely to experience positive caregiving than those from White backgrounds [41].

Within the context of mental health, caregivers supporting persons with mental problems also present significant challenges [31,32]. The extant literature has reported caregivers’ experiences among the general population in accessing mental health services [42,43,44]. It is evident that caregivers’ mental health literacy is limited; there is inadequate consumer and caregivers’ readiness for services, and mental healthcare professionals often misinterpret legal provisions for confidentiality and privacy relative to accessing mental health services for individuals in need of care [43]. Moreover, caregivers experience challenges in the protection of persons experiencing mental disorders, management of challenging behavior, over-reliance on religious and faith healers, financial barriers, and transportation difficulties [32]. In particular, they experience mental health issues [45,46,47,48,49,50,51,52], financial hardship [45], stigma [45,51], isolation [53], and disruption in their work [54,55,56,57]. For instance, a qualitative study [37] investigated the experiences of young care givers who assisted their family members with mental health disorders, and revealed that while these individuals felt a cultural obligation to support their relatives, they also encountered substantial barriers to education and stigma associated with caregiving. Furthermore, caregivers experience an increased burden and emotional distress [33]. However, the nuanced experiences faced by caregivers supporting deaf persons with mental health disorders have rarely been researched and so their needs and concerns are not adequately addressed in existing mental health policies [11,28].

The above situation call for the development of more formal training and support systems tailored to young caregivers. The existing literature remains limited in scope, with a particular gap on the experiences of caregivers of deaf persons with mental problems. It is imperative to generate data on the perspectives of caregivers of deaf persons with mental health conditions to complement existing studies so as to gain a comprehensive understanding of the unique experiences—opportunities and challenges—of these carers while providing mental health support in the context of Ghana. This understanding is crucial to inform policy and practice to improve mental health care in Ghana.

### Caregiving and Deaf Persons in Ghana

Ghana has made significant strides in the provision of mental health care with the formulation of Mental Health Act 846, which backed the creation of the Mental Health Authority [58]. The Ministry of Health supervises the delivery of mental health services [57] which is categorizes into three tiers: tertiary, secondary, and primary. Tertiary care encompasses teaching hospitals and specialized health facilities, while secondary care is provided through regional hospitals. Primary care consists of community and district health facilities. Each tier employs health professionals dedicated to assisting individuals with mental health challenges [57]. However, access to professional mental health services is still limited to many people compelling some people to opt for traditional methods such as traditional healers, priests, and churches for treatment [57,59,60]. Inadequate government funding, cost of treatment, shortage of mental health professionals, inadequate support services, and mental health stigma are the major factors limiting access to mental health care in Ghana [59,61,62].

It is estimated that out of 2,098,138 individuals with disabilities, 470,737 with some form of deafness are living in Ghana [63]. These deaf individuals form a community who are involved in activism led by the Ghana National Association of the Deaf [11]. The deaf community utilizes Ghanaian Sign Language, a shared mode of communication among themselves. Despite ongoing advocacy for the recognition of Ghanaian Sign Language as an official language, its practical application remains limited. Consequently, essential services such as healthcare, including mental health care, education, and transportation remain inaccessible to the deaf community [64,65,66,67,68]. Studies indicate that the deaf community has limited knowledge about the onset, causes, and effects of mental health on their development [11]. Although existing studies on access to mental health services for deaf individuals in Ghana is scant, there is a notable absence in the literature addressing the experiences of caregivers who support deaf persons with mental health disorders.

The existing literature on caregivers has predominantly concentrated on the experiences of parents raising children with deafness and other developmental disabilities [69,70]. It is noteworthy that mothers often assume the role of primary caregivers for children with hearing loss [69,71]. A qualitative study [69] highlighted various negative experiences reported by these mothers, including a lack of knowledge regarding appropriate caregiving strategies, challenges in effectively training their deaf children, familial tension, and difficulties with identifying suitable educational institutions for their deaf children. Similar challenges have been documented by other parents raising children with a range of disabilities such as cerebral palsy [72,73], visual impairments [74,75], intellectual disabilities [76], and autism [77,78] in Ghana and other contexts. The difficulties encountered by parents of children with deafness and other disabilities raise crucial questions regarding the strategies employed by caregivers supporting deaf individuals who have been diagnosed with mental health disorders.

Generally, the present mental health systems in Ghana remain underdeveloped [59,79,80], prompting ongoing discussions regarding necessary reforms and the creation of accessible mental health services for all. Integrating deaf persons’ caregivers’ perspectives into mental health research is vital to bridging the existing knowledge gaps and enriching scholarly understanding. Caregivers of deaf people with mental health disorders are likely to face unique challenges shaped by socioeconomic, cultural, and relational factors, which differ across contexts. Exploring their lived experiences will inform policies and interventions, enhancing well-being for caregivers and care recipients. Knowing and addressing stigmas and cultural nuances will also foster societal acceptance and equitable care. Significantly, the incorporation of these caregivers’ voices aligns with participatory approaches, advancing person-centered care, and designing culturally appropriate frameworks. The current study, therefore, attempted to understand the experiences of caregivers supporting deaf persons with mental health disorders. The study was guided by the following research questions: (a) how do caregivers of deaf persons with mental health problem conceptualize mental health? (b) what support systems are available to enhance the well-being of caregivers supporting deaf persons with mental health disorders? (c) in what ways has the experience of caregiving for a deaf person with mental health concerns influenced family relationships? and (d) what challenges do the caregivers encounter in their roles as carers of deaf persons with mental health problems?

## 2. Materials and Methods

### 2.1. Study Participants

The participant in this study comprised caregivers who provide support to deaf adults with mental health disorders in Ghana. This study is a part of a larger project which sought to understand awareness of mental health among deaf persons in Ghana [11,25]. To facilitate culturally sensitive and appropriate research practices, the research team collaborated with the Ghana National Association of the Deaf (GNAD). This partnership was essential to ensure effective engagement with the deaf community throughout the research process. Recognizing the importance of comprehensive representation, the deaf community advocated for the inclusion of participants from both of the two sectors (i.e., northern and southern) of Ghana. As a result, the study included two administrative regions from each sector to enhance representativeness. The northern sector was represented by the Northern and Upper West regions, while the southern sector was represented by the Central and Greater Accra regions.

Eleven caregivers supporting deaf persons with mental health disorders (represented by pseudonyms P1, P2, and P3 from the various regions) were purposively sampled taking into account the study’s objectives, the sample’s specificity, established theories, the depth and quality of dialogue, and the analytical strategy applied (i.e., the information power principle) [81]. Specifically, the intensity purposive sampling technique was used to facilitate the selection of participants capable of offering comprehensive insights and knowledge pertinent to the studied topic [82]. The research team engaged with health facilities across the four regions to connect with caregivers of deaf persons accessing mental health services within these institutions. In several instances, health facility staff contacted potential participants on behalf of the research team, considering those who expressed interest in participating in the study.

Similarly, invitations were sent to members of GNAD branches across the four regions to help locate the caregivers within their organization who were dealing with mental health disorders. The members of the association identified some caregivers who were then invited to participate in this study. Participants were recruited based on the following inclusion criteria: (a) a family member providing care to a deaf adult who had been diagnosed with a mental health problem; (b) a caregiver rendering an unpaid caregiving service; (c) a caregiver caring for the deaf person who had been diagnosed with a mental health problem for at least three years (justified by the caregiver’s nuanced, context-specific, and sustained experience over the period); (d) the caregiver had in-depth information about the condition of the deaf person; and (e) was willing to consent to participate in this study. While the study aimed to include any individual meeting the above conditions, incidentally, it was observed that most caregivers were women, and were also the biological mothers of the deaf adults living with the mental health disorders. However, one participant was identified as a sister and another as a brother to the deaf individual. They were all hearing individuals (see Table 1 for details).

### 2.2. Instrument

The study employed an exploratory descriptive design to capture and present the voices of caregivers of deaf adults diagnosed with mental problems. A semi-structured interview guide was developed for data collection. The interview format was chosen to encourage participants to express their experiences and insights freely. Amplificatory questions were asked where needed. A review of the existing literature on caregivers’ experiences [45,46,48,49,51] informed the development the interview guide. Some of the questions included: “How did you recognize that your deaf relative was experiencing mental health disorders”, “Can you share how you sought help for your deaf relative with mental health concerns”, “Can you share some of the training you have received to support your deaf relative with mental problems”, “how has the mental condition of your deaf relative affected your family”, and “Please share some of the challenges you encountered in caring for your deaf relative with mental problems”. The instrument was shared with the Ghana health services and deaf community for review before it was distributed for data collection.

### 2.3. Procedure

The study and its protocols received approval from the Institutional Review Committee at Ghana Health Service (GHS-ERC 026/09/21). Following this, health facilities in the region and GNAD were contacted to help identify potential participants who met the preset inclusion criteria. Both health facilities and GNAD liaised with potential participants and the details of those who consented to participate in this study were shared with the research team. Initially, a total of 19 potential participants were identified; however, only 11 were available for face-to-face interviews. The interviews were conducted by trained graduate students working as research assistants with experience in qualitative data collection. A number of potential participants were unreachable on the scheduled dates for data collection, while others chose to withdraw from participation without providing reasons. Non-participation by some caregivers may stem from mistrust of researchers, fear of emotional distress during discussions, or concerns about the potential misuse of shared information. These factors highlighted the need for trust-building, emotional sensitivity, and strict confidentiality during the research process.

The data were collected between January and June 2022, with the duration of the interviews ranging from 32 min to 70 min. The interviews were conducted in either English or the Twi language, which is the widely spoken local language in Ghana. The interviews were recorded with the consent of the participants. Prior to interviews, the participants were briefed about the study objectives and the importance of their contributions. They were assured about their anonymity in and the use of the data for the research. It is important to note that participants were not financially compensated for their involvement in the study. All participants signed an informed consent form prior to engaging in the research.

### 2.4. Data Analysis

The data were transcribed verbatim by the trained research assistants. Texts were reviewed by a language editor to ensure clarity and coherence before they were used in this study. Subsequently, one participant from each of the regions was invited to engage in discussions regarding the content of the transcribed data, as well as to highlight key insights from the interviews conducted. The data were analyzed using thematic analysis following these steps: reading, coding, mapping and categorizing, thematizing, and writing the findings sections [83].

Initially, the first author, along with two research assistants, read one set of interview data and highlighted the phrases amenable to coding. Consecutive meetings were held in which they discussed and refined these phrases into a cohesive coding frame, which was subsequently employed by the first author to code the remaining ten interview transcripts. Next, the first author categorized the data by mapping similarities between the interview excepts. Common descriptors were assigned to groups of similar ideas to facilitate analysis. Thereafter, the first author debriefed the research team about their progress before moving to the next step. Additionally, the descriptors generated from this process were tabulated according to the themes emerging from the research questions. Relevant quotes corresponding to the descriptors were extracted and compiled into a new Microsoft Word document to facilitate the development of the results section. Finally, all participants were given the opportunity to review the results section, and their agreement was obtained regarding the content presented therein, ensuring stakeholder validation in the reporting of the current study.

## 3. Results

### 3.1. Knowledge on Mental Health Problem

A significant number of caregivers exhibited a lack of awareness of the specific mental health diagnoses of their wards. Some of the caregivers said they “were only told he is mentally unstable but nothing else again” (P1, Central). Despite this lack of diagnostic clarity, caregivers were able to articulate observed behaviors in their wards, describing them as “reserved” and “see things that we don’t see, showing us things we cannot see” (P2, Upper West). Notably, only one caregiver identified her child’s condition as schizophrenia (P1, Northern), highlighting a gap in in the knowledge of their wards’ conditions probably due to inadequate communication between the caregivers mental health professionals.

Caregivers reported that alterations in the behavior of their wards served as critical indicators for identifying underlying conditions. A caregiver from the Central Region, for example, said “I realized a change in behavior and the way he used to do things around here. Everything he was doing was not normal to us” (P1, Central). Similarly, a participant from the Upper West Region expressed concerns about her child’s responsiveness, noting “He was not adequately responding to various assessment methods we employed to gauge his well-being and showed limited social interaction with others. That was the moment I recognized there was an issue” (P1, Upper West). These observations underscore f caregivers’ insights in the early detection of developmental or behavioral difficulties.

When caregivers were asked to describe how they felt about their wards’ behavior, two themes emerge: worrying or indifference. While those whose wards were aggressive were worried, the opposite was observed for some of those whose wards were reserved. Participant P2 from Central said her son “it is hard! He screams sometimes and also the notable one is getting up even from bed at odd hours and roaming in town to the extent that we would have to dispatch a search party to find him” (P2, Central). Two other caregivers whose child exhibited aggressive behavioral pattern reported the following:

“He always likes to roam about even at odd times like at midnight, but as to the specific thing that is causing him to do so, we have not been told, even though we took him to the hospital all the time. This is worrying.” (P1, Central)

“Hmmm, I think now is better unlike before. When he first had this condition, he was a bit wild and would roam about all the time. And also, since he has a measure of visual impairment, sometimes, if he fails to recognize what he has used to identify our house, he will pass by it and roam all round unless he gets help. Sometimes, after taking medicine, he is fine but by the next day, he will be making so much noise and if care is not taken, you might be tempted to give him an overdose so that he won’t disturb again.” (P2, Central)

Some of the caregivers demonstrated concerned as exemplified in sentiments expressed by these individuals. One caregiver remarked that “she is always reserved and doesn’t socialize with other people. Not even the siblings in the house. It was not like that when she was growing up” (P1, Upper West Region). This observation highlights a perceived decline in social engagement, prompting apprehension regarding the child’s development. Similarly, another caregiver described her ward’s condition, stating “…ok now, but just that she doesn’t mind anybody. She always stays by herself” (P2, Northern Region). These comments underscore caregivers’ worries about the social isolation of deaf persons diagnosed with mental disorders. Conversely, some caregivers exhibited a more relaxed attitude towards their wards’ reserved behavior. For instance, one stated “her behavior doesn’t cause him any worry. I am not really worried …well he is cool and does not do anything that brings extra worry to me” (P3, Central Region). This perspective suggests that some caregivers may not perceive reserved behavior as problematic, suggesting lack of knowledge, indifference, or variability in the interpretation of social withdrawal among different caregivers.

All the participants reported that they had sought professional care when they noticed the change in the behavior of their wards by sending then to the hospital. Some of the hospitals where the caregivers sent their wards to were Tamale Central Hospital, the University of Cape Coast Hospital, Ankaful Psychiatric Hospital, Mamprobi Polyclinic, Ewim Polyclinic in Cape Coast, and Wa Municipal Hospital. This suggests some of the caregivers were proactive in addressing behavioral changes in their wards using professional institutions.

In examining the impact of hospital treatment on the health conditions of deaf persons diagnosed with mental health disorders, responses among participants revealed a spectrum of experiences. While some participants reported minimal or no improvement in their wards, others noted some degree of progress. For instance, Participant P1 from the Central Region stated “For now, it is okay; he has been improving,” indicating a positive trajectory in his child’s health status. Conversely, a participant from the Northern Region expressed a contrasting experience, stating “Her condition is not better,” suggesting a lack of noticeable improvement. Furthermore, another participant from the Central Region shared her personal experience, providing more insights into the varied outcomes of treatment among the participants:

“Well, they take care of him like any other person but the situation has not really changed, just that we have been given new medications recently so we are waiting to see if that will bring any change but the previous ones, nothing at all.” (P2, Central)

### 3.2. Available Mental Health Support Services and Training

Some participants reported that they received support, while others reported a lack of assistance. Among those who received supports, they mainly came from the Department of Social Welfare, and were typically in the form of cash and/or counseling, often provided only once. Analysis of the responses suggests the supports they received was inadequate, and so some desired additional support as exemplified in the comment of one caregiver: “Yes, support is needed, but we only had help from the Department of Social Welfare a long time ago, and that is the only support we can recall” (P2, Central).

The caregivers in this study reported a significant lack of training in managing their wards with mental health disorder, which has caused a deficiency in their behavioral management skills. Consequently, many caregivers were inert in their approach, often allowing wards to exhibit uncontrolled behaviors. One parent articulated this sentiment, stating that: “I have no training, so we just do what we can and allow him to behave as he wants. Sometimes, we let him do as he wants, and he is okay. We leave him as such then later on he becomes calm” (P2, Central). Another caregiver expressed frustration regarding the absence of support, noting “I have not received any training. Just that he [the child with mental health disorders] is always admitted when we take him to the hospital and no other thing was done for me nor for him” (P2, Northern Region). These statements depict the critical need for structured training and support for caregivers of deaf people with mental health disorders to enhance their ability to effectively manage behaviors and improve outcomes for both deaf individuals with mental health disorders and their families.

Some caregivers reported that insufficient training in behavior management significantly hindered their ability to effectively care for their wards with mental health disorders. This gap in knowledge prompted a call for targeted training programs to enhance caregivers’ competencies in managing the specific needs of the deaf persons with mental health disorders under their care. A participant from the Central Region highlighted this need, stating that, “we need some knowledge about how to handle such issues [behavior challenges of deaf persons with mental health disorders]” (P2, Central Region). This sentiment was echoed by a participant from the Upper West Region, who emphasized that “training on how to cater for them [deaf persons with mental health disorders], patience, understanding, etc. is needed to properly manage the child” (P2, Upper West Region). Such findings underscore the critical role that professional development plays in improving caregiving practices and outcomes for persons with specialized needs.

Caregivers have emphasized the significance of providing training for deaf persons with mental health challenges to make them self-reliant. For instance, some participants indicated that vocational training in areas such as agriculture or carpentry can enhance their economic independence. A caregiver articulated this perspective by stating that “I think knowledge on how to take care of themselves is vital because, despite having a mental illness, they may still be capable of engaging in activities such as farming or carpentry” (P1, Central). Although this training has been proposed for their wards, it will indirectly benefit the caregivers by relieving them of their financial burden.

### 3.3. Psychosocial and Relationships with Families, Neighbours and Health Workers

The findings indicated that there were no psychosocial effects on the caregiver arising from caring for their wards with mental health disorders. The majority of the participants (*n* = 9) reported that they lived a ‘normal’ life as before, even though some wished that their wards with mental health disorders would relate to others better than they were doing. A caregiver commented that she lived a “normal life. Just that I wish she [her daughter with mental health disorder] could relate well with me and other people” (P1, Upper West), while another one said “I live a normal life. I don’t feel anything different about him. He lives normal life just like any other person” (Upper West). Two parents, however, had a contrary experience because they said their personal well-being had been affected. One of them said it affected her “badly, and sometimes I am unable to sleep” (P2), while the other said she was not happy because “he [son with mental health disorders] can’t work and that is very bad” (P3).

At the family level, participants indicated the existence of positive relational dynamics among family members. The presence of the child did not adversely affect trust, support they received from family members, respect, and stability in their marriages. This was exemplified by the willingness of family members to provide support and care for the child experiencing mental health challenges. For instance, Participant P1 from the Central Region stated: “We are all on good terms and relate well with each other, to the extent that when I am not present, they come around to offer support.” This demonstrates a strong familial bond that fostered a supportive environment for caring for mental health patients. The findings also indicated that the mental health challenges experienced by deaf people did not significantly impact marital relationships of the carers. For instance, Caregiver P3 from the Central Region remarked “it never affected my marriage, but now my husband is deceased,” while P2, also from the Central Region, stated that, “everything [regarding the marriage] is going on well.” This good relationship also existed between caregivers and their friends, healthcare workers, and the public. Caregivers explained that their friends and neighbors were on good terms with them and supported them when they needed support. Healthcare workers, according to the participants, exhibited good behavior towards them, and “there is nothing to worry about when it comes to them [neighbors] too (P1, Central). Healthcare workers “treat us well especially since I always visit the hospital” (P3, Central). Another said the following:

“Well, from the start, when I used to go with him to the hospital, they behaved well. I think even now that I ask others to take him to the hospital, I have not heard anything bad from them, meaning that everything has been good.” (P1, Central)

### 3.4. Challenges in Caring for People with Mental Health Disorders

#### 3.4.1. Economic Activities

The majority of the caregivers (*n* = 7) identified as self-employed individuals whose occupations included tailoring, bread baking, selling, and operating drinking spots. However, a subset of caregivers reported being unable to work due to health-related issues. In terms of the impact of their wards’ conditions on their professional activities, the majority of participants indicated that there were no adverse effects on their employment. For example, a participant said “I do think from time to time, but it has no negative impact on my business though” (P1, Northern). One participant, however, explained that she was “not working now but it really affected my job at first” (P3, Central). Another commented as follows:

“Yes, it [the child’s condition] really affects me. For instance, just yesterday I had to take him to the hospital and sometimes when he cannot be found, I stop all my work and search for him. It really bothers me a lot.” (P2, Central)

The majority of participants (*n* = 7) similarly indicated that the conditions of children with mental health disorders did not significantly impact on their financial status. For instance, one participant remarked “there is really no financial burden on me in caring for her” (P1, Upper West). Another one echoed this, saying “apart from his basic needs, there is no financial implication” (P2, Upper West). These responses align with previous statements regarding the effects of their wards’ conditions on their economic activities. However, there appears to be a discrepancy concerning their requests for support. A few participants, however, reported experiencing financial effects due to the necessity of allocating proceeds from their businesses to care for their wards with mental health disorders. One participant noted “yes, it has affected my finances, as I used it to take care of him, which wouldn’t be the case if he wasn’t in this condition.” (P2, Central).

Some caregivers reported that the cost of medical treatment was high, and this affected them, but that they obtained some relief by subscribing onto the National Health Insurance Scheme. However, they had to sometimes buy drugs that were not covered by the scheme or were not available in the hospital. For example, a caregiver said “first I was paying but now I don’t pay anything because of the insurance” while another caregiver indicated that they registered for the insurance so anytime “we visited the hospital, part of the expenses is catered for by insurance. We only buy the medicines that are not at the facility” (P2, Upper West). Some participants also complained that their wards would not allow them to take them to the hospital, probably due to stigma: “Medication is free under health insurance, but he doesn’t allow us to take him there [mental hospital]” (P2, Upper West).

#### 3.4.2. Public Attitude and Stigma

The public attitude towards deaf people with mental health disorders, according to the caregivers in the study, was positive. The participants expressed that the public was sympathetic towards deaf people with mental health conditions because they were considered as vulnerable. However, some caregivers indicated that the severity and nature of the mental health condition influenced public attitude. For example, people tended to show a more positive attitude towards those who are less/not aggressive than to those who are aggressive. One participant articulated that “well in our community here since he is not wild, they tend to treat him well, but I cannot tell of other places” (Northern Region P2). Another participant shared her experience in the quotation below.

“Well, what I can attest to is what happens in our community here. They are not hostile towards my son, and they all relate well with him. They know such people are vulnerable and they cannot do as they do so most often than not, they assist him when they come into contact with him.” (P1)

The perceptions among caregivers regarding mental health stigma in the context of deaf individuals reveal dichotomous viewpoints. While some thought that deaf people with mental health disorders experienced stigma, others did not think so. Caregivers who recognized stigmatization attributed it to the concept of “multiple disability.” For example, people “sometimes laugh at deaf people thinking they are mad” (P3, Central), people “look down on them [deaf people with mental health problems] because they have double conditions” (P1, Upper West), and they “tagged and labelled deaf people with mental health disorders because they are deaf and at the same time mentally ill” (P2, Northern).

However, as stated above, some participants thought deaf people with mental health disorders were not stigmatized. One participant explained “well, like I said earlier, I have not seen others being stigmatized but in my son’s case, I think because he was born and raised here, the stigma is not seen nor felt but rather, we see that people really empathize with us” (P1). This seems to suggest that the person was not stigmatized because of familiarity.

The relationship between deafness and societal attitudes towards individuals with mental health disorders was examined in this study, revealing a significant nuance in perceptions. Contrary to the responses of some participants, the findings indicate that deafness itself does not inherently influence societal attitudes. A majority of caregivers (*n* = 7) reported a generally positive societal perception of deaf individuals diagnosed with mental health challenges. For instance, one participant from the Upper West Region commented “for my daughter, people like her and always try playing and socializing with her, but she doesn’t engage. So, I would assert that societal attitudes towards her are positive. Deaf individuals are not treated differently” (P1, Upper West). Additionally, another participant noted “there is no influence since people regard and stigmatize them in the same way as hearing individuals with mental health disorders” (P1, Northern). These insights suggest that, while individual experiences may vary, the overarching societal attitude appears to lack a differential approach based solely on deafness.

Some participants reported that they would not entertain any negative attitude towards their wards and would fight anyone who mistreated them. This has probably influenced the positive attitude of neighbors towards deaf people with mental health disorders. A parent said the following:

“Here no one dare treats my child anyhow. For I will just come and fight with you so because of that I have not seen people treat my son badly and even with the other gentleman, I know in our area, they are not hostile to him.” (P3, Central)

However, a few caregivers said that deafness could worsen societal attitudes towards deaf people. Deafness “makes the situation worse since they have multiple disabilities, people pity them and do not want to be associated with them.” (P2, Northern).

## 4. Discussion

The findings showed that the participants were only able to detect mental health problems based on the actions of their family members with mental issues. Interview excerpts indicated that many participants struggled to detect the onset of mental problems among deaf individuals. This observation corroborates previous studies which reported the limited knowledge of mental health among caregivers and the general population [84,85]. Mental health problems and associated behavioral manifestations can be divided broadly into internalizing and externalizing behaviors. In this study, the participants could only identify mental health disorders based on externalizing behavior. However, most mental health conditions, such as anxiety, depression, and stress, typically originate internally before manifesting as externalizing behaviors, such as physical aggression, as outlined by the study participants. This finding underscores the possibility that a significant number of deaf persons with mental problems remain unidentified or undetected by their families. In the Ghanaian context, the findings reported in this study may not be surprising, considering the overall low level of public education about mental health, which has contributed to a significant risk of undiagnosed conditions within the population [57,79,80]. As part of efforts to promote an inclusive society, it is imperative for policymakers to engage with the deaf community to enhance awareness of the signs of mental health disorders, which could be integrated into mental health training curricula.

An unexpected finding of the study was the participants’ claims to pursue professional health services for deaf persons experiencing mental health concerns. Upon observing unusual behavior, participants reported sending deaf persons to hospital for support. In the Ghanaian context, seeking professional help for mental health problem is unusual, as many individuals tend to prefer traditional healing practices offered by priests and religious leaders [31,32,33]. In the same way, the parents of children with disabilities usually visit traditional priests and religious leaders for healing purposes [57,59,60]. This finding aligns with the role of caregivers, which indicates that they often seek medical assistance or accompany their family members with mental health challenges to healthcare facilities [69,74,86]. However, it is plausible that the caregivers may have previously tried traditional outlets and found them ineffective when it comes to healing disability and mental health disorders. This may have swayed them to seek professional support for the deaf persons diagnosed with mental health disorders. Nonetheless, the participants’ inclination to pursue professional mental health services is a promising development. This trend could be leveraged by the government through the enhancement of health facilities to ensure the provision of comprehensive mental health support services for all individuals in need.

Another insightful finding that emerged from this study was the negligible impact of mental health disorders on the participants. Almost all respondents reported that the mental health disorders of deaf persons did not significantly affect them psychologically or disrupt family unity or bonding. This finding stands in contrast to other studies that have documented the psychological ramifications of caregiving for mental health disorders on caregivers [86,87,88,89,90]. Moreover, the findings regarding the effect of caring for deaf persons with mental health disorders on economic, social, and psychological well-being were unexpected. Generally, it is anticipated that caring for a person with a disability, especially one with a mental health problem, would have adverse consequences on parents due the difficulty combining caregiving and working, the cost of medical expenses, the hostile attitude of neighbors, and managing the behavioral issues of the child [32,76,86,91,92]. However, most caregivers in the study did not experience any of these consequences. This is probably because the caregivers were supporting persons with mental health disorders who were adults and most of them were probably independent before they had mental health problems. It is also noteworthy that the participants may have developed resilience against the backdrop of the challenges they faced raising deaf individuals diagnosed with mental health disorders as reported elsewhere [40]. This aspect warrants further investigation in future research to provide deeper insights into the resilience exhibited by caregivers of deaf individuals with mental health concerns.

While there were positive experiences, some challenges were recounted by the study participants. For instance, a lack of training and irregular support to parents were highlighted by the study participants. Many said they neither received financial support nor training to care for deaf persons with mental health conditions. However, financial assistance was often limited to a one-time benefit, with no expectation of ongoing support. This finding is somewhat consistent with previous studies reporting the inadequacy of financial assistance and training offered to caregivers supporting persons with mental problems [76,84,93,94]. Additionally, reports indicate that individuals experiencing mental health disorders often do not receive financial support from the state [95]. This finding may not be surprising because persons with disabilities usually lack necessary formal assistance from the state [95,96,97,98,99]. Similarly, parents raising deaf children did not receive formal training to enhance their caregiving capabilities [70,76,100]. Based on this, the responses of the study’s participants were not unexpected. In Ghana, the welfare and needs of the caregivers of persons with mental health problems are yet to be considered in public discourse or policy. This finding is probably a wakeup call to health policymakers to consider developing social protection and training programs for all caregivers, including those supporting deaf persons with mental health disorders.

The financial implications of mental health rehabilitation were highlighted by study participants. For example, participants reported that the condition of deaf persons diagnosed with mental health disorders did not affect their job or income. However, they indicated that the cost of mental health rehabilitation was very high, exerting an adverse effect on their income. This finding aligns with the existing literature, which has consistently indicated financial strain in accessing mental health services [101,102]. Additionally, the literature has documented that the costs associated with disability and mental health rehabilitation in Ghana [95,98,99] and similar contexts [103,104] are very high, thereby restricting access for individuals without adequate financial support. In Ghana, the National Health Insurance Scheme (NHIS) supports primary healthcare only. Unfortunately, disability and mental health rehabilitation are not considered as primary healthcare. Consequently, individuals grappling with mental health challenges are often compelled to incur out-of-pocket expenses for necessary treatments. Prior research has advocated for the incorporation of disability and mental health rehabilitation services within the framework of primary healthcare, thereby enabling coverage under the National Health Insurance system (references [11,105,106,107]. The findings of this study further bolster the argument for policymakers to consider the integration of mental health rehabilitation into primary healthcare services under the NHIS.

This study also found a favorable public attitude and low stigma towards deaf persons experiencing mental health disorders. This was unexpected as the literature has shown that caregivers of individuals with mental health disorders are usually victimized and subjected to discrimination by the general populace [53,54,55,56]. Likewise, in the Ghanaian context, individuals living with mental health disorders typically encounter significant stigma and social discrimination [59,61,62]. One possible explanation for this unexpected outcome is that caregivers provided their perspectives on the experiences of adult deaf persons diagnosed with mental health disorders without the latter having the opportunity to share their own narratives. Previous studies have indicated that deaf persons may possess limited knowledge about mental health disorders [11,25,30,108].

It is possible that the accounts of deaf persons experiencing mental health disorders could differ from what was reported by their caregivers. While the findings of this study appear promising, future research should incorporate the voices of deaf individuals who have recovered from mental health disorders to gain a deeper understanding of their personal experiences.

### 4.1. Implication for Practice

The findings of this study have significant implications for mental health rehabilitation among the deaf population in Ghana. Firstly, it is essential for health policymakers to actively engage with the deaf community and their families regarding mental health concerns. This engagement can inform the development of a comprehensive training manual aimed at enhancing the capabilities of caregivers who support deaf individuals. Secondly, practitioners are encouraged to establish structured programs specifically designed for the parents of deaf persons diagnosed with mental health disorders. Such programs should be conducted over an extended period to facilitate a deeper understanding of the conditions affecting deaf persons diagnosed with mental health disorders, as well as to impart best management practices. Lastly, as highlighted in this study, there is a pressing need for policymakers to include mental health rehabilitation within the scope of conditions covered under primary healthcare policies. This inclusion would play a crucial role in alleviating financial barriers, thereby increasing accessibility to mental health services for deaf individuals. Furthermore, it is recommended that future studies explore potential reasons for the significant resilience of caregivers raising deaf persons diagnosed with mental health disorders in the study context. This knowledge will inform appropriate interventions in similar contexts.

### 4.2. Study Limitations

The current study is subject to several limitations. First, the study drew on caregivers only, without integrating the voices of persons living with mental health disorders. Future studies should consider incorporating the perspectives of those who have recovered from mental health disorders. Second, the study did not capture the insights of healthcare providers or members of society regarding deaf individuals with mental health challenges, which could have enriched the findings. Third, the participant sample was drawn from only four out of the sixteen regions, limiting the generalizability of the results. Notably, while some regions share similarities in economic activities and demographics which may allow for some parallels to be drawn, the findings are not necessarily representative of the entire country. Fourth, like all qualitative studies, the small sample size restricts the generalizability of the findings to the broader population of caregivers supporting deaf individuals with mental health disorders. Future research could benefit from a quantitative design to facilitate comparisons across diverse demographics. Moreover, it was beyond the scope of this study to evaluate the extent and severity of the mental conditions of deaf persons with mental health disorders. However, since they were recruited by health service providers as well as the GNAD, there is certainty that the deaf persons were living with a form of mental health disorder. Furthermore, participants who took part in this study were caring for deaf adults and thus responses cannot be extrapolated for deaf children who are living with mental problems. Future research could specifically recruit the caregivers of deaf children diagnosed with mental health conditions to address this gap in the literature.

## 5. Conclusions

The current study contributes to the existing literature on caregivers of individuals with mental health disorders by specifically exploring the experiences of those supporting deaf individuals living with mental health disorders. The findings did not report participants’ suffering from psychological distress, financial hardship, familial tensions, and discord and discrimination from society at large. Nevertheless, the caregivers expressed concerns regarding their limited knowledge of mental health disorders, the unavailability of relevant training opportunities, the prohibitive costs associated with such training, and inadequate social protection mechanisms. While these findings are encouraging, they highlight persistent gaps in the provision of mental health services for deaf individuals. Overall, this study is a pioneering effort that enriches the scholarly discourse on caregiving practices for deaf individuals with mental health concerns, particularly within the context of Ghana, where ongoing debates regarding mental health reforms are taking place.

## Figures and Tables

**Table 1 ijerph-22-00144-t001:** Profile of the study participants.

	Category of Participant	Region	Gender	Age	Relation	Mar. Stat.	Occupation	Age of Deaf Person
1	P1	Central	Female	61	Mother	Married	Fish monger	44
2	P2	Central	Female	68	Mother	Single	Fish monger	39
3	P3	Central	Female	49	Sister	Widow	Trader	34
4	P1	Northern Region	Male	45	Brother	Single	Unemployed	45
5	P2	Northern Region	Female	56	Mother	Separated	Unemployed	35
6	P3	Northern Region	Female	59	Mother	Widow	Unemployed	33
7	P1	Upper West	Female	58	Mother	Single	Unemployed	41
8	P2	Upper West	Female	57	Mother	Married	Trading	32
9	P3	Upper West	Female	54	Mother	Married	Farmer	33
10	P1	Greater Accra	Female	58	Mother	Divorced	Trading	37
11	P2	Greater Accra	Female	63	Mother	Divorced	Trading	31

## Data Availability

The data sets are not publicly available as they contain information that could potentially identify participants, but are available from the corresponding author upon reasonable request and with relevant ethical approval.

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
