# Peer review of "Caregivers’ Experience of Supporting Deaf Adults with Mental Health Disorders in Ghana"

_ijerph, 2025, doi:10.3390/ijerph22020144_

Round 1

Reviewer 1 Report

Comments and Suggestions for Authors

Thank you for allowing us to read about research in this area of the confluence between deafness and mental health disorders. As this is an exploratory study and because your self-criticism meets my main doubts, I think you will be able to be published.

I would like you to consider some variables in future work that were not taken into account here. Namely, the way deaf people communicate, the socioeconomic level of the families to which they belong, their marriage/professional status, and their age.

Thank you again for the opportunity.

Author Response

Please find attached response to comments

Reviewer 2 Report

Comments and Suggestions for Authors

I would like to thank the authors for their submission and allowing me to review their work.

This is an interesting study on an important topic. However, I would be grateful if you could add further explanations and changes on the following points:

1) ABSTACT: Page 1, Line 27

I suggest specifying the mean age (± standard deviation) and the sex distribution of the study population.

2) ABSTRACT: Page 1, line 27

I suggest specifying whether the caregivers themselves are primarily family members or professionals.

3) ABSTRACT: Page 1, line 28

What specific types of mental disorders have been studied?

4) INTRODUCTION: Page 2, line 56

I suggest providing more specific examples of the barriers caregivers face in accessing mental health services.

5) MATERIALS AND METHODS: Page 5, line 181

I suggest providing a rationale for choosing the specific duration of three years.

6) MATERIALS AND METHODS: Page 5, line 210

I suggest adding a brief discussion of potential reasons for non-participation.

7) RESULTS: Page 8, line 358; Pag. 9, line 391; Page 9, line 403; Page 11, line 456

How many caregivers out of the total sample held these views? I suggest adding the numbers or percentages to make the findings more robust.

8) DICUSSION:

The study demonstrates that caregivers did not report significant psychological distress, financial hardship, or familial tensions. This finding is intriguing, and it would be useful to explore potential reasons for this resilience among caregivers of deaf individuals with mental health concerns.

Author Response

see attached document

Reviewer 3 Report

Comments and Suggestions for Authors

This study reports on the experiences of 11 adult midddle-aged to older adult caregiver of 11 middle-aged deaf adults in Ghana with mental health issues. 1) The authors never say much about deafness as a feature of these individuals, their needs or their caregiver relationships. 2) Throughout the paper, the deaf persons are referred to as the children of the caregiver or as their wards but their status as adults and the length of the caregiver relationship are never discussed as factors. 3)The representativeness and/or diversity or other indicators of appropriateness of the sample are discussed little: readers from outside of Ghana will not know the significance of the specific regions referenced without further explication, and the table 1 indication of "cat" and "P1, P2, P3 etc." were not clear to me from the text. 4) the introduction and methods sections do not justify and detail the questions asked to respondents, and 5) I think the most striking part of your findings was that caregivers and their communities had more often than not found a way to normalize, accept and even appreciate the deaf person with mental health issues and I don't think the introduction or the discussion adequately contrast this finding with the more negative experiences in European-centered/individually-oriented/competitive cultures.... I think the opportunity to speak about mutual adjustment over time and cultural values on community were missed.; and 6) I really appreciated your recommendations for practice---a more structured set of educational tools and other supports for famies with deaf children facing mental health or other chronic diseases---and it was not at all clear how the study findings and interpretations led to this conclusion.

This is a small qualitative study with a very unusual sample and you have not gone nearly far enough to explain what you learned that is generalizable and why your findings have larger implications. Because there is little information about mental health services and supports for deaf people in Ghana that has appeared in international health journals, including these discussions and even strengthening them will be an important asset for this paper.

Author Response

see attached response to the comments

Round 2

Reviewer 2 Report

Comments and Suggestions for Authors

The Authors have clarified my doubts and improved the manuscript.

Reviewer 3 Report

Comments and Suggestions for Authors

I have no further suggestions for the authors. I am really impressed by their work in taking the comments from the 3 reviewers and adding significant new content and context to the paper and clarifying research goals, methods and findings.